# Connecting Optimization and Regularization Paths

**Arun Sai Suggala**
Carnegie Mellon University
Pittsburgh, PA 15213
asuggala@cs.cmu.edu

**Adarsh Prasad**
Carnegie Mellon University
Pittsburgh, PA 15213
adarshp@cs.cmu.edu

**Pradeep Ravikumar**
Carnegie Mellon University
Pittsburgh, PA 15213
pradeepr@cs.cmu.edu

## Abstract

We study the implicit regularization properties of optimization techniques by explicitly connecting their optimization paths to the regularization paths of "corresponding" regularized problems. This surprising connection shows that iterates of optimization techniques such as gradient descent and mirror descent are *pointwise* close to solutions of appropriately regularized objectives. While such a tight connection between optimization and regularization is of independent intellectual interest, it also has important implications for machine learning: we can port results from regularized estimators to optimization, and vice versa. We investigate one key consequence, that borrows from the well-studied analysis of regularized estimators, to then obtain tight excess risk bounds of the iterates generated by optimization techniques.

## 1 Introduction

With the recent success of optimization techniques in training over-parametrized deep neural networks, there has been a growing interest in understanding the implicit regularization properties of various optimization techniques. Consequently, a line of work has focused on characterizing the implicit biases of global optimum reached by various optimization algorithms. For example, Gunasekar et al. [2017] consider the problem of matrix factorization and show that gradient descent (GD) on un-regularized objective converges to the minimum nuclear norm solution. Soudry et al. [2017] study gradient descent on un-regularized logistic regression and show that when the data is linearly separable, gradient descent converges to a max-margin solution. Gunasekar et al. [2018] generalized the results of Soudry et al. [2017] and study the limit behavior of the iterates of general optimization techniques when the data is linearly separable.

Another line of work has focused on studying the implicit regularization properties of *early stopping* various optimization algorithms, which is a widely used technique in neural network training. These works show that early stopping the iterative optimization of an empirical problem performs a form of implicit regularization. Yao et al. [2007] focus on non-parametric regression in reproducing kernel Hilbert spaces and provide theoretical justification for early stopping. In a similar setting, Raskutti et al. [2014] show that early stopping gradient descent on least squares objective achieves similar risk bounds as the corresponding regularized problem, also called ridge regression. Hardt et al. [2015], Rosasco and Villa [2015] study the implicit regularization properties of early stopping stochastic gradient descent (SGD). All these results show that early stopping achieves similar performance as optimizing the corresponding regularized objective.

Furthermore, several recent works suggest that there could be a much deeper connection between the iterates generated by optimization techniques on un-regularized objectives (*optimization path*) and minimizers of corresponding regularized objectives (*regularization path*), than the performance similarity observed in the early stopping literature. Friedman and Popescu [2003] empirically observe that for linear regression, the optimization and regularization paths are very similar to each other. Rosset et al. [2004a] show that under certain conditions on the problem, the path traced by coordinate

descent or boosting is similar to the regularization path of $L_1$ constrained problem. In a related work Neu and Rosasco [2018] consider the problem of linear least squares regression and show that the iterates produced by GD on least squares objective are related to the solutions of ridge regression. Specifically, for any given regularization parameter of ridge regression, Neu and Rosasco [2018] show that there exists a weighing scheme for GD iterates that is exactly equal to the ridge solution.

In this work, we take a step towards understanding the deeper connection between the two paths by explicitly connecting the optimization path to the regularization path of the corresponding regularized problem. Our results explicitly show that the sequence of iterates produced by iterative optimization techniques such as gradient descent, mirror descent on strongly convex functions, lie *pointwise* close to the regularization path of a corresponding regularized objective. This surprising connection allows us to transfer insights from regularization to optimization and vice-versa. We expect that our work will lead to a new class of results in both fields that explicitly draw upon this connection.

In this paper, we focus on a particular consequence of our connection: we derive excess risk bounds of the iterates of optimization techniques. There has been a huge line of work in the fields of machine learning and statistics on understanding the risk bounds of regularized problems [Negahban et al., 2009, Hsu et al., 2012]. We utilize these results to derive excess risk bounds of iterates of optimization techniques.

Recently, there has been a line of work studying the excess risk of iterates of optimization techniques. Yao et al. [2007], Raskutti et al. [2014] focus on non-parametric regression in a reproducing kernel Hilbert space and derive excess risk bounds of iterates of gradient descent. Wei et al. [2017] extend these results to a broad class of loss functions. In the context of finite dimensional spaces, Hardt et al. [2015], Chen et al. [2018] use the notion of algorithmic stability, which was introduced by Bousquet and Elisseeff [2002], to derive bounds on the expected excess risk of iterates of various methods. Our technique for deriving excess risk bounds can be viewed as an alternative to stability and has the advantage that we can make use of the existing results on the statistical properties of regularized problems. Moreover, this approach has the potential to obtain much tighter bounds than stability and we stress that any improvement in the analysis of regularized estimation will directly translate to a tighter bound for the corresponding optimization problem.

The main contributions of the paper are as follows. For strongly convex and smooth objectives, we explicitly connect the optimization path of GD and regularization path of $L_2^2$ penalized objective. We further extend these results to Mirror Descent with strongly convex and smooth divergences. We use these connections to derive the excess risk of iterates of GD. For convex objectives, we show that the connection need not hold in general. However, for the problem of classification with separable data, we show that for losses with exponentially decaying tails, the optimization path of GD is close to the regularization path of the corresponding regularized objective.

## 2 Strongly Convex Loss

In this section we explicitly connect the optimization path of GD and regularization path of $L_2^2$ penalized objective on strongly convex and smooth functions. Let $f : \mathbb{R}^p \to \mathbb{R}$ be a twice differentiable function which is strongly convex and smooth with parameters $m, M > 0$. In this work we mainly focus on continuous time GD (that is, GD with infinitesimally small step size). We define the *optimization path* of GD on $f(\theta)$, started at $\theta_0$, as the trajectory followed by GD iterates, which is given by the following Ordinary Differential Equation (ODE)

$$\dot{\theta}(t) := \frac{d}{dt}\theta(t) = -\nabla f(\theta(t)), \quad \theta(0) = \theta_0.$$

We now relate the above optimization path to the *regularization path* of the corresponding $L_2^2$ penalized objective, which is defined as the 1-dimensional path of optimal solutions of the following regularized objective, obtained as we vary $\nu$ between $[0, \infty)$

$$\underline{\theta}(\nu) = \operatorname*{argmin}_{\theta} f(\theta) + \frac{1}{2\nu}\|\theta - \theta_0\|_2^2. \tag{1}$$

The following Lemma bounds the distance between the optimization and regularization paths.

**Theorem 1.** *Let $\hat{\theta}$ be the minimizer of $f(\theta)$. Let $\kappa = m/M$ and let $c = \frac{2\kappa}{1+\kappa}$. Moreover, let the regularization penalty $\nu$ and time $t$ be related through the relation $\nu(t) = \frac{1}{cm}\left(e^{cMt} - 1\right)$. Suppose*

*GD is started at $\theta_0$. Then*

$$\|\theta(t) - \underline{\theta}(\nu(t))\|_2 \leq \frac{\|\nabla f(\theta_0)\|_2}{m}\left(e^{-mt} + \frac{c}{1 - c - e^{cMt}}\right).$$

Note that when $\kappa = 1$ the upper bound in the above Theorem is equal to 0, thus showing that both the paths are exactly the same. To get a sense of quality of the bound, we compare it with a simple triangle inequality based bound, where we derive an upper bound for $\|\theta(t) - \underline{\theta}(\nu(t))\|_2$ by first bounding $\|\theta(t) - \hat{\theta}\|_2$, $\|\underline{\theta}(\nu(t)) - \hat{\theta}\|_2$ and then using a triangle inequality.

**Theorem 2** (Weak Bound). *Consider the similar setting as in Theorem 1. Let $\nu(t) = \frac{1}{cm}\left(e^{cMt} - 1\right)$. Then*

$$
\begin{aligned}
\|\theta(t) - \underline{\theta}(\nu(t))\|_2 \quad &\leq \quad \|\theta(t) - \hat{\theta}\|_2 + \|\underline{\theta}(\nu(t)) - \hat{\theta}\|_2 \\
&\leq \quad \frac{\|\nabla f(\theta_0)\|_2}{m}\left(e^{-mt} + \frac{c}{1 - c + e^{cMt}}\right).
\end{aligned}
$$

The above Theorem gives us $O(e^{-mt} + e^{-Mt})$ upper bound for the distance $\|\theta(t) - \underline{\theta}(\nu(t))\|_2$. Whereas, Theorem 1 gives us $O(e^{-mt} - e^{-Mt})$ upper bound, which is strictly better than Theorem 2. Moreover, for small $t$, the bound in Theorem 2 is much weaker than the bound in Theorem 1. As we show later, the tighter bound in Theorem 1 helps us obtain tight generalization bounds and early stopping rules for the iterates of GD.

By choosing a different relation $\nu(t)$, one can obtain a different connection and a different upper bound for the distance between optimization and regularization paths. In Appendix A.1 we consider different choices for $\nu(t)$ and obtain different bounds. We believe the bounds in Theorem 1 and Appendix A.1 can be further improved by choosing an "optimal" $\nu(t)$.

## 2.1 Extension to Mirror Descent

In this section we provide an extension of Theorem 1 to Mirror descent. Before we proceed, we briefly review mirror descent. For a complete review of properties of mirror descent and Bregman divergences see [Banerjee et al., 2005, Bubeck et al., 2015]. Let $\phi$ be a continuously differentiable Legendre function defined on $\mathbb{R}^p$. Moreover let $\phi$ be $\alpha-$strongly convex w.r.t a reference norm $\|.\|$

$$\phi(\theta_2) - \phi(\theta_1) - \langle\nabla\phi(\theta_1), \theta_2 - \theta_1\rangle \geq \frac{\alpha}{2}\|\theta_2 - \theta_1\|^2.$$

Then the Bregman divergence $D_\phi$ induced by $\phi$ is defined as $D_\phi(\theta_2, \theta_1) = \phi(\theta_2) - \phi(\theta_1) - \langle\nabla\phi(\theta_1), \theta_2 - \theta_1\rangle$.

**Mirror Descent (MD).** Suppose we want to minimize a convex function $f(\theta)$ over $\mathbb{R}^p$. Then Mirror Descent with divergence $D_\phi$ uses the following update rule to estimate the minimizer

$$\theta^{t+1} = \underset{\theta}{\operatorname{argmin}} f(\theta^t) + \langle\nabla f(\theta^t), \theta - \theta^t\rangle + \frac{1}{\eta_t}D_\phi(\theta, \theta^t).$$

Solving the above equation gives us the following update rule: $\nabla\phi(\theta^{t+1}) = \nabla\phi(\theta^t) - \eta_t\nabla f(\theta^t)$. The continuous time dynamics of MD, started at $\theta_0$, is given by the following ODE

$$\dot{\theta}(t) = -\nabla^2\phi(\theta(t))^{-1}\nabla f(\theta(t)), \quad \theta(0) = \theta_0.$$

$D_\phi$**-Regularization.** We connect the optimization path of MD to the regularization path of the following regularized problem

$$\underline{\theta}(\nu) = \underset{\theta}{\operatorname{argmin}} f(\theta) + \frac{1}{\nu}D_\phi(\theta, \theta_0),$$

where $\theta_0$ is some point in $\Theta$. The solution $\underline{\theta}(\nu)$ satisfies the following continuous time dynamics

$$\dot{\underline{\theta}}(\nu) := \frac{d}{d\nu}\underline{\theta}(\nu) = -\left[\nu\nabla^2 f(\underline{\theta}(\nu)) + \nabla^2\phi(\underline{\theta}(\nu))\right]^{-1}\nabla f(\underline{\theta}(\nu)).$$

We now show that the optimization path of mirror descent with $D_\phi$ divergence, started at $\theta_0$, is closely related to the regularization path of the corresponding $D_\phi$-regularized objective. Our analysis is similar to the analysis of GD.

**Theorem 3.** *Let $\hat\theta$ be the minimizer of $f(\theta)$. Suppose $f$ is $m$ strongly convex, $M$ smooth and $\phi$ is $\alpha$ strongly convex w.r.t euclidean norm. Moreover, suppose $\phi$ is $\beta$ smooth w.r.t euclidean norm, in the following balls around $\theta_0$ and $\hat\theta$*

$$\{\theta : D_\phi(\theta, \theta_0) \leq D_\phi(\hat\theta, \theta_0)\} \cup \{\theta : D_\phi(\hat\theta, \theta) \leq D_\phi(\hat\theta, \theta_0)\}.$$

*Let the regularization penalty $\nu$ and time $t$ be related through the relation $\nu(t) = \frac{\beta}{cm}\left(e^{cMt/\alpha} - 1\right)$, where $c = 2\left(1 + \frac{\beta}{\alpha}\frac{1}{\kappa}\right)^{-1}$ and $\kappa = m/M$. If MD is started at $\theta_0$, then*

$$\|\theta(t) - \underline{\theta}(\nu(t))\|_2 \leq \frac{\beta}{\alpha}\frac{\|\nabla f(\theta_0)\|_2}{m}\left(e^{-mt/\beta} + \frac{c}{1 - c - e^{cMt/\alpha}}\right).$$

Note that when $\beta = \alpha$, we retrieve the bounds of GD in Theorem 1. An example of a divergence which satisfies the assumption of strong convexity and smoothness of $\phi$ over $\mathbb{R}^d$ is the Mahalanobis distance which is a divergence induced by the function $\phi(x) = x^T A x$, for some positive definite matrix $A$. However, we note that many popular divergences such as KL-divergence do not satisfy the smoothness condition over the entire space $\mathbb{R}^d$. For such divergences, $\beta$ depends on the distance between the starting point $\theta_0$ and the minimizer $\hat\theta$ and their location in the space.

## 3 Consequences for Excess Risk of GD Iterates

In this section we utilize the connection between optimization and regularization paths derived in Theorem 1 to provide excess risk bounds of GD iterates. To this end, we first derive excess risk bounds of the solutions of the regularized problem and then combine it with the result from Theorem 1 to obtain the excess risk bounds of GD iterates.

### 3.1 General Analysis

In this section we provide a general statistical analysis of the solution of the regularized problem in Equation (1), for general statistical learning problems. Suppose we are given $n$ i.i.d samples $D_n = \{x_i\}_{i=1}^n$, where $x_i \in \mathcal{X}$, drawn from a distribution $P$. Let $\ell : \mathbb{R}^p \times \mathcal{X} \to \mathbb{R}$ be a loss function that assigns a cost $\ell(\theta, x)$ for an observation $x$. Define the risk $R(\theta)$ as $R(\theta) = \mathbb{E}_{X \sim P}\left[\ell(\theta, X)\right]$ and let $\theta^*$ be the minimizer of risk $R(\theta)$. Given samples $D_n$, our goal is to use $D_n$ to obtain an estimate $\hat\theta$ that has low excess risk: $R(\hat\theta) - \min_\theta R(\theta)$. Let $R_n(\theta)$ denote the empirical risk, which is defined as $R_n(\theta) = \frac{1}{n}\sum_{i=1}^n \ell(\theta, x_i)$. We consider the following regularized problem for estimating a $\theta$ with low excess risk

$$\min_\theta R_n(\theta) + \frac{1}{2\nu}\|\theta\|_2^2. \tag{2}$$

The following theorem bounds the parameter estimation error of the minimizer of the above problem.

**Theorem 4.** *Suppose the empirical risk $R_n(\theta)$ is $m$ strongly convex and $M$ smooth. Consider the regularized problem in Equation (2). Suppose the regularization penalty $\nu$ satisfies $1/\nu \geq 2\frac{\|\nabla R_n(\theta^*)\|_2}{\|\theta^*\|_2}$. Then the optimal solution $\underline{\theta}(\nu)$ satisfies*

$$\|\underline{\theta}(\nu) - \theta^*\|_2 \leq \frac{3}{m\nu}\|\theta^*\|.$$

Using the above result and the result from Theorem 1 we now bound the parameter error of the iterates of continuous time GD.

**Corollary 5.** *Suppose the conditions of Theorem 1 are satisfied. Moreover, let $t$ satisfy $t \leq \frac{1}{cM}\log\left(1 + \frac{cm\|\theta^*\|}{2\|\nabla R_n(\theta^*)\|_2}\right)$, where $c = \frac{2m}{m+M}$. Then $\theta(t)$ satisfies the following error bound*

$$\|\theta(t) - \theta^*\|_2 \leq \frac{\|\nabla R_n(\theta_0)\|_2}{m}\left(e^{-mt} + \frac{c}{1 - c - e^{cMt}}\right) + \frac{3}{c}\frac{e^{-cMt}}{1 - e^{-cMt}}\|\theta^*\|.$$

Note that the above results provide deterministic error bounds for a particular choice of $\nu, t$. The random quantities $m, M, \|\nabla R_n(\theta^*)\|_2$ need to be bounded to instantiate the above result for specific learning problems.

Let $\bar{m}, \bar{M}$ be the strong convexity and smoothness parameters of the population risk $R(\theta)$. Using standard tools from empirical process theory, under certain regularity conditions on the distribution $P$ and $R(\theta)$, one can show that $\|\nabla R_n(\theta^*)\|_2, \|\nabla R_n(0)\|_2$ scale as $O(\sqrt{\frac{p}{n}}), O(\sqrt{\frac{p}{n}} + \|\theta^*\|)$ and $m, M$ are close to $\bar{m}, \bar{M}$ with high probability. Substituting these in Corollary 5 gives us the following bound for $\|\theta(t) - \theta^*\|_2$ at $t = \frac{1}{\bar{c}\bar{M}} \log\left(1 + \frac{\bar{c}\bar{m}\|\theta^*\|}{2}\sqrt{\frac{n}{p}}\right)$

$$\|\theta(t) - \theta^*\|_2 = O\left(\left(e^{-\bar{m}t} - \bar{c}e^{-\bar{M}t}\right)\|\theta^*\| + \sqrt{\frac{p}{n}}\right),$$

where $\bar{c} = \frac{2\bar{m}}{\bar{m}+\bar{M}}$. When $\bar{m} = \bar{M}$, the above bound shows that at $t = \frac{1}{\bar{M}}\log\left(1 + \frac{\bar{M}\|\theta^*\|}{2}\sqrt{\frac{n}{p}}\right)$, we achieve the standard parametric error rate $O\left(\sqrt{\frac{p}{n}}\right)$.

We note that the above bound can be improved with a tighter analysis of the regularized problem. In the next section, we consider the problem of linear regression and show how a better understanding of the regularized problem, coupled with our connection, helps us obtain a tighter parameter estimation error bound for the iterates of GD.

## 3.2 Tighter Analysis for Linear Regression

Recall that in linear regression we observe paired samples $\{(x_1, y_1), \ldots (x_n, y_n)\}$, where each $(x_i, y_i) \in \mathbb{R}^p \times \mathbb{R}$. The distribution of $y$ conditioned on the covariates $x$ is specified by the following linear model: $y = \langle x, \theta^* \rangle + w$, where $w$ is drawn from a zero-mean distribution with bounded variance. In this work we assume that noise has a normal distribution with variance $\sigma^2$; that is, $w \sim \mathcal{N}(0, \sigma^2)$. The goal in linear regression is to learn a linear map $x \to \langle \theta, x \rangle$ with low risk $R(\theta) = \mathbb{E}\left[(y - \langle \theta, x \rangle)^2\right]$. The empirical risk $R_n(\theta)$ is given by

$$R_n(\theta) = \frac{1}{2n}\sum_{i=1}^{n}(y_i - \langle \theta, x_i \rangle)^2 = \frac{1}{2n}\|y - X\theta\|_2^2, \tag{3}$$

where $X = [x_1, x_2, \ldots x_n]^T \in \mathbb{R}^{n \times p}$ is the matrix of covariates. Let $w = y - X\theta^*$ be the noise vector and $\widehat{\Sigma}$ be the empirical covariance matrix. The regularized problem (2) corresponding to the least squares risk defined above is called ridge regression problem. Ridge regression has been well studied and analyzed in the literature of machine learning and statistics. We now provide the following result from Hsu et al. [2012] which obtains tight upper bounds on the parameter estimation error of the solution of ridge regression.

**Theorem 6.** *Suppose the covariate vector $x$ has a normal distribution with mean $0$ and covariance matrix $\Sigma$. Then there exists constants $c_1, c_2 > 0$ that depend on $\Sigma$, such that for $n \geq c_1 p \log p$, the solution of ridge regression $\underline{\theta}(\nu)$ satisfies the following error bound with probability at least $1 - 1/p^2$*

$$\|\underline{\theta}(\nu) - \theta^*\|_2^2 \leq \frac{1}{\kappa}\left[\underbrace{\left\|\left((\Sigma + \frac{1}{\nu}I)^{-1}\Sigma - I\right)\theta^*\right\|^2}_{Bias} + \underbrace{\frac{\sigma^2}{n}\sum_{i=1}^{p}\left(\frac{\nu\lambda_i}{1+\nu\lambda_i}\right)^2}_{Variance}\right],$$

*where $\lambda_i$ is the $i^{th}$ largest eigenvalue of $\Sigma$, where $\kappa = \lambda_p/\lambda_1$.*

We now use the above bound to obtain error rates for optimization path of GD. For the sake of simplicity and to gain insights into the bound we consider the special case of identity covariance matrix.

**Corollary 7.** *Suppose the covariate vector $x$ has a normal distribution with mean $0$ and identity covariance matrix. Then there exists a constant $c_1 > 0$ such that for $n \geq c_1 p \log p$, the iterates of continuous time GD satisfy the following bound with probability at least $1 - 1/p^2$*

$$\|\theta(t) - \theta^*\|_2 \leq \frac{10}{9}\left(e^{-9t/10} - \frac{9}{10e^{99t/100}-1}\right)\left(\|\theta^*\| + \sigma\sqrt{\frac{p}{n}}\right) + \frac{1}{1+\nu(t)}\sqrt{\|\theta^*\|^2 + \nu(t)^2\sigma^2\frac{p}{n}},$$

*where $\nu(t) = \frac{100}{81}\left(e^{99t/100} - 1\right)$. Further, at $t = \frac{100}{99}\log\left(1 + \frac{81}{100}\frac{\|\theta^*\|^2}{\sigma^2}\frac{n}{p}\right)$, the iterate $\theta(t)$ satisfies*

$$\|\theta(t) - \theta^*\|_2^2 \leq (1 + \epsilon)\left[\frac{\|\theta^*\|^2}{\|\theta^*\|^2 + \frac{\sigma^2 p}{n}}\right]\frac{\sigma^2 p}{n},$$

*where $\epsilon$ is a positive constant less than $0.1$.*

Note that the above bound provides an early stopping rule for GD on linear regression, and the resulting rate, especially in the high SNR regime where $\sigma$ is high, can be better than the $\frac{\sigma^2 p}{n}$ rate obtained by running GD until convergence.

**Comparison with Stability.** Hardt et al. [2015], Chen et al. [2018] used stability as a technique to provide expected excess risk bounds of iterates generated by an iterative optimization algorithm. We note that in the setting of strong convexity, existing stability based approaches impose a much stronger condition of strong convexity on $R_n(\theta)$. Specifically, they require the loss function $\ell(\theta, x)$ to be strongly convex in $\theta$ at each $x \in \mathcal{X}$. For example, this condition never holds for linear regression with dimension $p > 1$. Under the assumption that the loss function $\ell(\theta, x)$ at each $x \in \mathcal{X}$ is $m$ strongly convex and $M$ smooth, stability gives us the following expected risk bounds for $\theta(t)$

$$\mathbb{E}\left[R(\theta(t)) - R(\theta^*)\right] \leq \frac{1}{n}\left(1 - e^{-2mt}\right) + e^{-2mt}.$$

Note that the above bound doesn't provide an early stopping rule and suggests that one has to run the algorithm until convergence for the best possible rates. Moreover, our approach can obtain high probability statements, whereas the above rates are in expectation.

**Comparison with VC, Rademacher complexity bounds.** Traditional techniques for bounding excess risk of iterates proceed by separately bounding the optimization error $\|\theta(t) - \hat{\theta}\|$, statistical error $\|\hat{\theta} - \theta^*\|$ and then using a simple triangle inequality to bound $\|\theta(t) - \theta^*\|$. Such a technique gives us rates of the form $O(e^{-mt} + \sigma\sqrt{\frac{p}{n}})$. These rates suggest that one should always run GD until the end to obtain best possible rates and can't predict optimal early stopping rules. Note that the bound in Corollary 7, $\|\theta(t) - \theta^*\|_2^2 \leq \underbrace{\left[\|\theta^*\|_2^2 / \left(\|\theta^*\|_2^2 + \sigma^2 p/n\right)\right]}_{\alpha} \frac{\sigma^2 p}{n}$, is much better than $O\left(\sigma^2 \frac{p}{n}\right)$

obtained using standard VC and Rademacher complexity bounds. This is especially true in the low SNR regime, where $\sigma$ is large and as a result $\alpha$ is low. Moreover, this rate is same as ridge regression rate. This shows that GD with early stopping can obtain similar rates as ridge regression; and rates which are tighter than those obtained via VC bounds and stability based risk bounds.

## 4 Convex Loss

Having studied the setting where $f(\cdot)$ is strongly convex, we next turn our attention to losses which are simply convex. As we show below, in the convex case, the connection between the two paths is more nuanced, and in particular, is problem specific.

Firstly, we derive a result which characterizes the end-point of the regularization path.

**Theorem 8.** *Let $f : \mathbb{R}^p \to \mathbb{R}$ be a convex function. Suppose $f$ has a minimizer. Let $\underline{\theta}(\nu)$ be the minimizer of*

$$f(\theta) + \frac{1}{2\nu}\|\theta - \theta_0\|_2^2.$$

*Then as $\nu \to \infty$, $\underline{\theta}(\nu)$ converges to the minimizer which is closest to $\theta_0$.*

We note that this result can be viewed as the regularization path analog of the result of Gunasekar et al. [2017], where it was shown that for matrix factorization, the optimization path of GD converges to the minimum Frobenius norm solution. Next, we present a simple counterexample which shows that in the convex regime, both regularization and optimization paths need not converge to the same point.

**Lemma 1.** *Consider the following function in 2D space $f : \mathbb{R} \times (-100, \infty) \mapsto \mathbb{R}$, $f(x, y) = \frac{(x+1)^2}{y+100}$. Suppose the continuous time gradient descent is initialized at $\theta_0 = (2, 1)$. Then, we have that*

$$\lim_{\nu \to \infty} \underline{\theta}(\nu) \neq \lim_{t \to \infty} \theta(t),$$

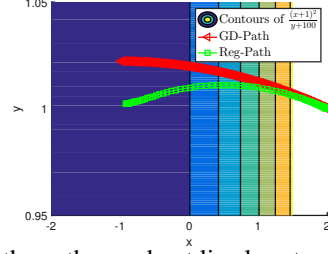

The above result shows that for general convex losses, both the paths need not lie close to each other, even as $t, \nu \to \infty$.

## 4.1 Classification Loss

In this section, we focus on classification losses and show that the optimization path of GD and the corresponding regularization path of $L_2^2$ penalized risk are close to each other. Commonly used losses in classification such as exponential, logistic loss are not strongly convex and moreover, when the data $D_n$ is separable, the risk doesn't admit a finite minimizer. In such cases, a more careful analysis is needed to bound the distance between the optimization and regularization paths.

Recent works by Ji and Telgarsky [2018], Soudry et al. [2017] study the behavior of gradient descent on un-regularized logistic regression and show that when the data is separable, GD converges to a max margin solution. In this section we first show that similar properties hold for the regularization path of $L_2^2$ regularized objectives. Recall that in classification we observe samples $D_n = \{(x_i, y_i)\}_{i=1}^n$, where each $(x_i, y_i) \in \mathbb{R}^p \times \{\pm 1\}$. Let $\ell(\theta, (x, y)) = \phi(yx^T\theta)$ be the loss at $(x, y)$. Consider the regularized problem in Equation (2). We first present the following useful result from Rosset et al. [2004b] which shows that when the data is linearly separable, as $\nu \to \infty$, the minimizer $\underline{\theta}(\nu)$ of (2) converges to a max-margin solution.

**Lemma 2.** *Assume the data $D_n$ is linearly separable; that is, $\exists \tilde{\theta}$ such that $\min_i y_i \left\langle x_i, \tilde{\theta} \right\rangle > 0$. Let $\phi(z)$ be a monotone non-increasing loss function. If $\exists T > 0$ (possibly $T = \infty$) such that:*

$$\lim_{t \to T} \frac{\phi(t(1 - \epsilon))}{\phi(t)} = \infty, \forall \epsilon > 0,$$

*then $\phi$ is a margin maximizing loss function in the sense that any convergence point of the normalized solutions $\frac{\theta(\nu)}{\|\theta(\nu)\|_2}$ of the regularized problem (2) as $\nu \to \infty$ is an $L_2$ margin maximizing separating hyper-plane. Consequently, if this margin-maximizing hyper-plane is unique, then the solutions converge to it*

$$\lim_{\nu \to \infty} \frac{\underline{\theta}(\nu)}{\|\underline{\theta}(\nu)\|_2} = \operatorname*{argmax}_{\|\theta\|_2 = 1} \left[ \min_i y_i \theta^T x_i \right].$$

The condition on $\phi$ in the above Lemma is satisfied by many popular loss functions such as logistic, exponential, squared hinge losses. Note that Lemma 2 is *asymptotic* in nature. Our first contribution is to derive a non-asymptotic version of this theorem. We focus on the exponential loss $\phi(z) = e^{-z}$, but our results can be generalized to other losses as well. Perhaps interestingly, our non-asymptotic bounds depend on the Lambert W(product-log) function [Corless et al., 1996], which has a long history of applications to instrument design [Ohayon and Ron, 2013] and statistical physics [Valluri et al., 2000].

**Theorem 9.** *Assume the data $D_n$ is linearly separable; that is, $\exists \tilde{\theta}$ such that $\min_i y_i \left\langle x_i, \tilde{\theta} \right\rangle = 1$. Let $\ell(\theta, (y, x)) = \exp(-\theta^T(yx))$ and let $\underline{\theta}(\nu)$ be the solution to the regularized problem in Equation (2). Then $\underline{\theta}(\nu)$ satisfies*

i) $R_n(\underline{\theta}(\nu)) \leq C_1 \frac{\mathcal{W}(\nu)}{\nu} = O\left( \frac{\log(\nu)}{\nu} \right)$, *where $\mathcal{W}(\cdot)$ is the Lambert W function.*

ii) $\|\underline{\theta}(\nu)\|_2 = \Theta(\log(\nu))$

iii) $\frac{\min_i y_i x_i^T \underline{\theta}(\nu)}{\|\underline{\theta}(\nu)\|_2} \geq 1 - \frac{\log \log(\nu)}{\log(\nu)} - \frac{\log n}{\log \nu}.$

As $\nu \to \infty$, the above Theorem shows that the $\underline{\theta}(\nu)$ converges to a max-margin solution, thus recovering the asymptotic result of Rosset et al. [2004b]. Moreover, our result shows that the minimizer of the regularized problem (2) converges to max-margin solution at a slow rate. In particular, the margin increases as $O(\frac{1}{\log \nu})$.

**Comparison with GD on $R_n(\theta)$.** Soudry et al. [2017] analyze gradient descent on exponential loss, with separable data and obtained similar bounds for the iterates of GD. Letting $\theta(t)$ be the iterate of GD at time $t$, they show that $R_n(\theta(t))$ goes down as $O(1/t)$, the margin converges as $O(1/\log t)$ and $\|\theta(t)\|_2$ increases as $\log t$. When combined with our result, this shows that the optimization and regularization paths are very close to each other.

**Theorem 10.** *Assume the data $D_n$ is linearly separable; that is, $\exists \tilde{\theta}$ such that $\min_i y_i \left\langle x_i, \tilde{\theta} \right\rangle = 1$. Let $\ell(\theta, (y, x)) = \exp(-\theta^T(yx))$. Suppose the regularization parameter $\nu$ and time $t$ are related as $\nu(t) = t$. Suppose GD is initialized at $0$. Then for any $t \geq 0$, we have*

$$\left| \min_{i \in [n]} \frac{y_i \left\langle x_i, \theta(t) \right\rangle}{\|\theta(t)\|_2} - \min_{i \in [n]} \frac{y_i \left\langle x_i, \underline{\theta}(\nu(t)) \right\rangle}{\|\underline{\theta}(\nu(t))\|_2} \right| \leq O\left( \frac{1}{\log t} \right).$$

# 5 Experiments

In this section, we conduct simulations to corroborate our theoretical findings.

## 5.1 Strongly Convex

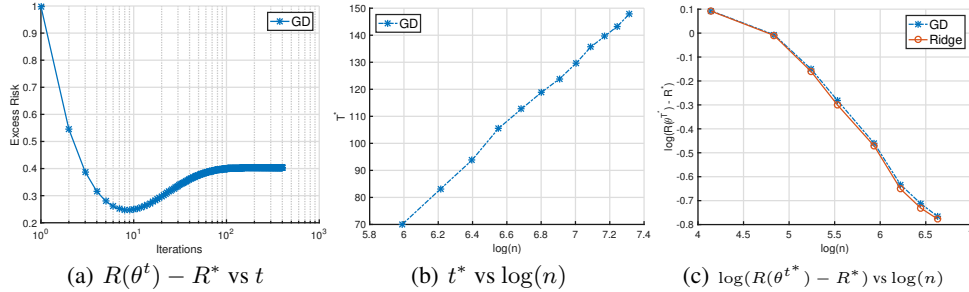

(a) $R(\theta^t) - R^*$ vs $t$      (b) $t^*$ vs $\log(n)$      (c) $\log(R(\theta^{t^*}) - R^*)$ vs $\log(n)$

Figure 1: Connecting GD and $L_2^2$-Penalization for Linear Regression

We use linear regression to empirically verify our results on connecting ridge-regression and gradient descent. We also corroborate our findings on excess risk and optimality of early-stopping rule for gradient descent.

**Setup.** We simulate a linear model by drawing the covariates from an isotropic gaussian $X \sim \mathcal{N}(0, \mathcal{I}_{p \times p})$ and the response $y|x \sim \mathcal{N}(\theta^{*T}x, \sigma^2)$ where $\theta^* = [1/\sqrt{p}, 1/\sqrt{p}, \ldots, 1/\sqrt{p}]^T$ and $\sigma^2 = 2$. We generate a sequence of iterates by GD with step size 0.01, and a corresponding sequence of solutions for the penalized estimation problem. We also study how the optimal iteration number $t^*$, which minimizes the excess risk, changes as we increase the number of samples for GD. In this case, we fix $p = 100$ and vary the samples $n$ from 100 to 1500. Similarly, we find the optimal penalization $\nu^*$ for each $n$. All results are reported after averaging over 50 trials.

**Results.** We report our results in Figure 1.

- As shown by our theory, excess risk bounds for GD on OLS are composed of two terms, one which increases with $t$ and the other which decreases with $t$. Hence, one expects the excess risk to first decreases, then increase before finally settling, which is corroborated by Figure 1(a).

- Figure 1(b) shows a logarithmic relationship between $t^*$ and $n$, thereby verifying our theoretical claims on $t^*$.

- Figure 1(c) shows that optimal risk for GD coincides with that of $L_2^2$-penalized estimation across different values of $n$.

## 5.2 Classification

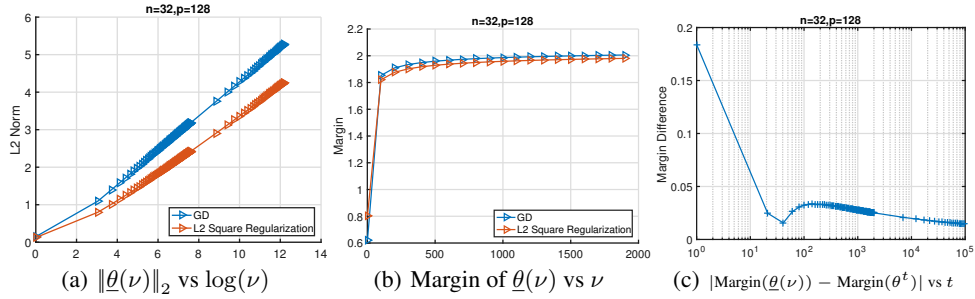

Figure 2: Connecting GD and $L_2^2$-Penalization for Logistic Regression

In this section, we corroborate our results which connect GD and $L_2^2$-regularization in the context of logistic regression on separable data. In particular, we corroborate our findings on parameter error, behavior of margin and the difference in margin for optimization and regularization paths.

**Setup.** We construct a classification dataset by drawing covariates $X$ from isotropic gaussian *i.e.* $X \sim \mathcal{N}(0, \mathcal{I}_p)$. We fix the true parameter $\theta^* = [\frac{1}{\sqrt{p}}, \frac{1}{\sqrt{p}}, \dots, \frac{1}{\sqrt{p}}]^T$. We fix the dimension $p = 128$ and the number of samples to $n = 32$. Note that our choice of $p, n$ ensures that the generated data is separable. We run GD with a step size $\eta = 0.123$ and construct corresponding points on the regularization path $(\nu(t) = \frac{t}{\eta})$ of the $L_2^2$ squared penalized objective.

**Results.** We report our results in Figure 2.

- Figure 2(a) shows the norm of the points on the optimization and regularization paths. As predicted by our theory, the norm increases at a logarithmic rate.

- Figure 2(b) plots the $L_2$-margin ($\frac{\min_i y_i \theta^T(x_i)}{\|\theta\|_2}$) for both the optimization and regularization path. The figure confirms our results that the margin increases with $\nu$.

- Although the margins of both the optimization and regularization paths increase, Figure 2(c) depicts that after a few initial iterations, the difference in the margin between $\theta^t$ and $\underline{\theta}(\nu)$ decreases.

## 6 Summary and Future Work

In this work, we studied the connections between the trajectory of the iterates of optimization techniques such as GD, Mirror Descent and regularization path of the corresponding regularized objective. For strongly convex functions our results show that both the paths are point-wise close. However, for general convex functions, our results show that both the paths need not be close to each other. For the popularly studied problem of classification with separable data, we showed that the optimization and regularization paths are close to each other.

We believe studying the connection between optimization and regularization paths has several advantages, with the key advantage being that it can be used to study the statistical properties of iterates generated by optimization techniques. We also believe that our results on strongly convex losses can be further improved to obtain tighter connections and better generalization bounds of the iterates.

An interesting direction for future work would be to see if similar connections hold for non-convex problems and specifically the optimization objectives that arise in deep learning. For convex losses, our current work focused on analyzing classification losses with separable data. It would be interesting to study the connection for general convex losses and identify the conditions on the loss function under which both the paths stay close to each other.

While our analysis in this paper focused on GD, it'd be interesting to study if similar connections hold for other non-stochastic methods such as steepest descent, accelerated GD, Newton's method and stochastic methods such as SGD.

# 7 Acknowledgement

We acknowledge the support of NSF via IIS-1149803, IIS-1664720, DMS-1264033. The authors are grateful to Suriya Gunasekar and anonymous reviewers for helpful comments on the paper.

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
