[Supplementary Material]

# A  Proof of Theorem 1

Before we prove the Theorem, we present an auxiliary Lemma, the proof of which can be found in Appendix F.

**Lemma 3.** *Let $\hat{\theta}$ be the minimizer of $f(\theta)$. Then for any $t, \nu \in [0, \infty)$, $\theta(t), \underline{\theta}(\nu)$ satisfy*

$$\|\nabla f(\theta(t))\|_2^2 \le e^{-2mt}\|\nabla f(\theta_0)\|_2^2$$

$$\|\nabla f(\underline{\theta}(\nu))\|_2^2 \le \frac{1}{(m\nu + 1)^2}\|\nabla f(\theta_0)\|_2^2.$$

We are now ready to prove our main Theorem. First note that the trajectory of $\underline{\theta}(\nu)$ is given by the following ODE

$$\dot{\underline{\theta}}(\nu) := \frac{d}{d\nu}\underline{\theta}(\nu) = -\left[\nu\nabla^2 f(\underline{\theta}(\nu)) + I\right]^{-1}\nabla f(\underline{\theta}(\nu)).$$

This follows from the observation that $\nabla f(\underline{\theta}(\nu)) + \frac{1}{\nu}(\underline{\theta}(\nu) - \theta_0) = 0$. To simplify the notation in the proof, we often suppress the dependence of $\nu$ on $t$ and just use $\nu$ instead of $\nu(t)$. We first derive an upper bound for $\frac{d}{dt}\|\theta(t) - \underline{\theta}(\nu)\|^2$. Let

$$H(\nu) = \left[\nu\nabla^2 f(\underline{\theta}(\nu)) + I\right]^{-1}, \kappa = \frac{m}{M}.$$

We have

$$\frac{d}{dt}\|\theta(t) - \underline{\theta}(\nu)\|^2 = 2\left\langle \dot{\theta}(t) - \frac{1}{\kappa}e^{cMt}\dot{\underline{\theta}}(\nu), \theta(t) - \underline{\theta}(\nu) \right\rangle \tag{4}$$

$$= -2\left\langle \nabla f(\theta(t)) - \frac{1}{\kappa}e^{cMt}H(\nu)\nabla f(\underline{\theta}(\nu)), \theta(t) - \underline{\theta}(\nu) \right\rangle$$

Let $G(\nu) = \frac{1}{\kappa}e^{cMt}H(\nu)$. Note that $G(\nu)$ is PSD whose the minimum and maximum eigenvalues satisfy

$$\sigma_{min}(G(\nu)) \ge \frac{c}{1 + (c\kappa - 1)e^{-cMt}},$$

$$\sigma_{max}(G(\nu)) \le \frac{c}{\kappa}\frac{1}{1 + (c-1)e^{-cMt}}.$$

We now obtain an upper bound for the RHS of Equation (4). We rewrite the RHS as

$$\frac{d}{dt}\|\theta(t) - \underline{\theta}(\nu)\|^2 = -2\underbrace{\langle\nabla f(\theta(t)) - \nabla f(\underline{\theta}(\nu)), \theta(t) - \underline{\theta}(\nu)\rangle}_{T_1} + 2\underbrace{\langle(G(\nu) - I)\nabla f(\underline{\theta}(\nu)), \theta(t) - \underline{\theta}(\nu)\rangle}_{T_2}.$$

$$\tag{5}$$

From strong convexity of $f$, we have the following lower bound for $T_1$

$$T_1 \ge m\|\theta(t) - \underline{\theta}(\nu)\|^2$$

From Lemma 3 we have the following upper bound for $T_2$

$$T_2 \le \|G(\nu) - I\|\|\nabla f(\underline{\theta}(\nu))\|\|\theta(t) - \underline{\theta}(\nu)\| \le \left(\frac{\|\nabla f(\theta_0)\|}{m\nu(t) + 1}\right)\|G(\nu) - I\|\|\theta(t) - \underline{\theta}(\nu)\|$$

Note that $\|G(\nu) - I\|$ can be upper bounded as

$$\begin{aligned}
\|G(\nu) - I\| &\le \max\{|1 - \sigma_{min}(G(\nu))|, |1 - \sigma_{max}(G(\nu))|\} \\
&\le \max\left\{\left|1 - \frac{c}{1+(c\kappa-1)e^{-cMt}}\right|, \frac{c}{\kappa}\frac{1}{1+(c-1)e^{-cMt}} - 1\right\} \\
&= \frac{c}{\kappa}\frac{1}{1+(c-1)e^{-cMt}} - 1,
\end{aligned}$$

where the last equality follows from our choice of $c$. Letting the above upper bound be $\sigma_G$, we get the following bound for $T_2$

$$T_2 \le \sigma_G\left(\frac{\|\nabla f(\theta_0)\|}{m\nu(t) + 1}\right)\|\theta(t) - \underline{\theta}(\nu)\|$$

Substituting these bounds in Equation (5) gives us

$$\frac{d}{dt}\|\theta(t) - \underline{\theta}(\nu)\|^2 \le -2m\|\theta(t) - \underline{\theta}(\nu)\|^2 + 2\sigma_G\left(\frac{\|\nabla f(\theta_0)\|}{m\nu(t) + 1}\right)\|\theta(t) - \underline{\theta}(\nu)\|.$$

We now solve the above ODE. Let $g(t) = \|\theta(t) - \underline{\theta}(\nu)\|$. The above equation can be rewritten as

$$\frac{d}{dt}g(t)^2 \le -2mg(t)^2 + \frac{2\sigma_G}{m\nu(t) + 1}\|\nabla f(\theta_0)\|g(t)$$

$$= -2mg(t)^2 + \frac{2\sigma_G}{\frac{1}{c}(e^{cMt} - 1) + 1}\|\nabla f(\theta_0)\|g(t)$$

$\frac{2\sigma_G}{\frac{1}{c}(e^{cMt}-1)+1}$ can be rewritten as

$$\frac{2\sigma_G}{\frac{1}{c}(e^{cMt} - 1) + 1} = \left(\frac{c}{\kappa} - 1 + (1-c)e^{-cMt}\right)\frac{2ce^{-cMt}}{(1 + (c-1)e^{-cMt})^2}.$$

This gives us the following upper bound for the ODE

$$\frac{d}{dt}g(t)^2 \leq -2mg(t)^2 + \left(\frac{c}{\kappa} - 1 + (1-c)e^{-cMt}\right)\frac{2ce^{-cMt}}{(1 + (c-1)e^{-cMt})^2}\|\nabla f(\theta_0)\|g(t)$$

Ignoring the trivial solution $g(t) = 0$, we can rewrite the above ODE as

$$\frac{d}{dt}g(t) \leq -mg(t) + \left(\frac{c}{\kappa} - 1 + (1-c)e^{-cMt}\right)\frac{ce^{-cMt}}{(1 + (c-1)e^{-cMt})^2}\|\nabla f(\theta_0)\|$$

Solving the ODE with the initial condition $g(0) = 0$, gives us the following upper bound for $g(t)$[1]

$$g(t) = \|\theta(t) - \underline{\theta}(\nu)\| \leq \frac{\|\nabla f(\theta_0)\|_2}{m}\left(e^{-mt} + \frac{c}{1 - c - e^{cMt}}\right).$$

## A.1  Other Relations

**Theorem 11.** *Let $\hat{\theta}$ be the minimizer of $f(\theta)$. Let $\kappa = m/M$ and let c be any constant such that $\frac{1+\kappa}{2} \leq c \leq 1$. Moreover, let the regularization penalty $\nu$ and time t be related through the relation $\nu(t) = \frac{1}{cM}\left(e^{cMt} - 1\right)$. Suppose GD is started at $\theta_0$. Then*

$$\|\theta(t) - \underline{\theta}(\nu(t))\|_2 \leq \frac{\|\nabla f(\theta_0)\|_2}{m}\left(e^{-mt} - \frac{\frac{c}{\kappa}}{e^{cMt} + \frac{c}{\kappa} - 1}\right).$$

**Proof of Theorem 11.**  The proof follows along similar lines as the proof of Theorem 1. We have

$$\frac{d}{dt}\|\theta(t) - \underline{\theta}(\nu)\|^2 = 2\left\langle\dot{\theta}(t) - e^{cMt}\underline{\dot{\theta}}(\nu), \theta(t) - \underline{\theta}(\nu)\right\rangle \tag{6}$$
$$= -2\left\langle\nabla f(\theta(t)) - e^{cMt}H(\nu)\nabla f(\underline{\theta}(\nu)), \theta(t) - \underline{\theta}(\nu)\right\rangle$$

Let $G(\nu) = e^{cMt}H(\nu)$. $G(\nu)$ is a PSD matrix whose the minimum and maximum eigenvalues satisfy

$$\sigma_{min}(G(\nu)) \geq \frac{c}{1 + (c-1)e^{-cMt}},$$

$$\sigma_{max}(G(\nu)) \leq \frac{c}{\kappa}\frac{1}{1 + (\frac{c}{\kappa} - 1)e^{-cMt}}.$$

We now obtain an upper bound for the RHS of Equation (6). We rewrite the RHS as

$$\frac{d}{dt}\|\theta(t) - \underline{\theta}(\nu)\|^2 = -2\underbrace{\langle\nabla f(\theta(t)) - \nabla f(\underline{\theta}(\nu)), \theta(t) - \underline{\theta}(\nu)\rangle}_{T_1} + 2\underbrace{\langle(G(\nu) - I)\nabla f(\underline{\theta}(\nu)), \theta(t) - \underline{\theta}(\nu)\rangle}_{T_2}.$$
$$\tag{7}$$

From strong convexity of $f$, we have the following lower bound for $T_1$

$$T_1 \geq m\|\theta(t) - \underline{\theta}(\nu)\|^2$$

From Lemma 3 we have the following upper bound for $T_2$

$$T_2 \leq \|G(\nu) - I\|\|\nabla f(\underline{\theta}(\nu))\|\|\theta(t) - \underline{\theta}(\nu)\| \leq \left(\frac{\|\nabla f(\theta_0)\|}{m\nu(t) + 1}\right)\|G(\nu) - I\|\|\theta(t) - \underline{\theta}(\nu)\|$$

$\|G(\nu) - I\|$ can be upper bounded as

$$\|G(\nu) - I\| \quad\leq\quad \max\{|1 - \sigma_{min}(G(\nu))|, |1 - \sigma_{max}(G(\nu))|\}$$
$$\leq\quad \max\left\{\left|1 - \frac{c}{1+(c-1)e^{-cMt}}\right|, \frac{c}{\kappa}\frac{1}{1+(\frac{c}{\kappa}-1)e^{-cMt}} - 1\right\}$$
$$=\quad \frac{c}{\kappa}\frac{1}{1+(\frac{c}{\kappa}-1)e^{-cMt}} - 1,$$

where the last equality follows from our choice of $c$ and some simple algebra. Letting the above upper bound be $\sigma_G$, we get the following bound for $T_2$

$$T_2 \leq \sigma_G \left( \frac{\|\nabla f(\theta_0)\|}{m\nu(t) + 1} \right) \|\theta(t) - \underline{\theta}(\nu)\|$$

Substituting these bounds in Equation (5) gives us

$$\frac{d}{dt} \|\theta(t) - \underline{\theta}(\nu)\|^2 \leq -2m\|\theta(t) - \underline{\theta}(\nu)\|^2 + 2\sigma_G \left( \frac{\|\nabla f(\theta_0)\|}{m\nu(t) + 1} \right) \|\theta(t) - \underline{\theta}(\nu)\|.$$

We now solve the above ODE. Let $g(t) = \|\theta(t) - \underline{\theta}(\nu)\|$. The above equation can be rewritten as

$$\frac{d}{dt} g(t)^2 \leq -2mg(t)^2 + \frac{2\sigma_G}{m\nu(t) + 1} \|\nabla f(\theta_0)\| g(t)$$

$$= -2mg(t)^2 + \frac{2\sigma_G}{\frac{\kappa}{c}(e^{cMt} - 1) + 1} \|\nabla f(\theta_0)\| g(t)$$

$\frac{\sigma_G}{\frac{\kappa}{c}(e^{cMt}-1)+1}$ can be rewritten as

$$\frac{\sigma_G}{\frac{\kappa}{c}(e^{cMt} - 1) + 1} = \left( \frac{c}{\kappa} - 1 + \left(1 - \frac{c}{\kappa}\right) e^{-cMt} \right) \frac{\frac{c}{\kappa} e^{-cMt}}{\left(1 + \left(\frac{c}{\kappa} - 1\right) e^{-cMt}\right)^2}.$$

This gives us the following upper bound for the ODE

$$\frac{d}{dt} g(t)^2 \leq -2mg(t)^2 + \left( \frac{c}{\kappa} - 1 + \left(1 - \frac{c}{\kappa}\right) e^{-cMt} \right) \frac{2\frac{c}{\kappa} e^{-cMt}}{\left(1 + \left(\frac{c}{\kappa} - 1\right) e^{-cMt}\right)^2} \|\nabla f(\theta_0)\| g(t)$$

Ignoring the trivial solution $g(t) = 0$, we can rewrite the above ODE as

$$\frac{d}{dt} g(t) \leq -mg(t) + \left( \frac{c}{\kappa} - 1 + \left(1 - \frac{c}{\kappa}\right) e^{-cMt} \right) \frac{\frac{c}{\kappa} e^{-cMt}}{\left(1 + \left(\frac{c}{\kappa} - 1\right) e^{-cMt}\right)^2} \|\nabla f(\theta_0)\|$$

Solving the ODE with the initial condition $g(0) = 0$, gives us the following upper bound for $g(t)$

$$g(t) = \|\theta(t) - \underline{\theta}(\nu)\| \leq \frac{\|\nabla f(\theta_0)\|_2}{m} \left( e^{-mt} - \frac{\frac{c}{\kappa}}{e^{cMt} + \frac{c}{\kappa} - 1} \right).$$

# B   Proof of Theorem 2

We first derive upper bounds for $\|\theta(t) - \hat{\theta}\|_2$, $\|\underline{\theta}(\nu(t)) - \hat{\theta}\|_2$ and use them to obtain an upper bound for $\|\theta(t) - \underline{\theta}(\nu(t))\|_2$.

Note that since $f$ is strongly convex, the following holds for any $\theta$

$$m\|\theta - \hat{\theta}\|_2 \leq \|\nabla f(\theta)\|_2.$$

From Lemma 3 we know that the following inequalities hold for any $t, \nu$

$$\|\nabla f(\theta(t))\|_2 \leq e^{-mt} \|\nabla f(\theta_0)\|_2, \quad \|\nabla f(\underline{\theta}(\nu))\|_2 \leq \frac{1}{(m\nu + 1)} \|\nabla f(\theta_0)\|_2.$$

Combining the results from Lemma 3 with the previous inequality, we get

$$\|\theta(t) - \hat{\theta}\|_2 \leq \frac{e^{-mt}}{m} \|\nabla f(\theta_0)\|_2, \quad \|\underline{\theta}(\nu(t)) - \hat{\theta}\|_2 \leq \frac{1}{m(m\nu(t) + 1)} \|\nabla f(\theta_0)\|_2$$

So we have the following upper bound for $\|\theta(t) - \underline{\theta}(\nu(t))\|_2$

$$\|\theta(t) - \underline{\theta}(\nu(t))\|_2 \leq \frac{1}{m} \left( e^{-mt} + \frac{1}{m\nu(t) + 1} \right) \|\nabla f(\theta_0)\|_2$$

$$\leq \frac{1}{m} \left( e^{-mt} + \frac{c}{1 - c + e^{cMt}} \right) \|\nabla f(\theta_0)\|_2.$$

# C  Proof of Theorem 3

The proof proceeds along the lines of Theorem 1. Before we prove the Theorem, we present an auxiliary Lemma, the proof of which can be found in Appendix F.

**Lemma 4.** *Let $\hat{\theta}$ be the minimizer of $f(\theta)$. Then for any $\nu \in [0, \infty)$, $\underline{\theta}(\nu)$ satisfies*

$$D_\phi(\underline{\theta}(\nu), \theta_0) \leq D_\phi(\hat{\theta}, \theta_0).$$

*For any $t \in [0, \infty)$, $\theta(t)$ satisfies*

$$D_\phi(\hat{\theta}, \theta(t)) \leq D_\phi(\hat{\theta}, \theta_0).$$

*This shows that the entire optimization and regularization paths lie in a region where $\phi$ is smooth with smoothness parameter $\beta$.*

**Lemma 5.** *Let $\hat{\theta}$ be the minimizer of $f(\theta)$. Then for any $\nu \in [0, \infty)$, $\underline{\theta}(\nu)$ satisfies*

$$\|\nabla f(\underline{\theta}(\nu))\|^2 \leq \left(\frac{\beta}{m\nu + \beta}\right)^2 \|\nabla f(\theta_0)\|^2.$$

We now prove our main Theorem. To simplify the notation, we often suppress the dependence of $\nu$ on $t$ and just use $\nu$ instead of $\nu(t)$. We first derive an upper bound for $\frac{d}{dt}\|\theta(t) - \underline{\theta}(\nu)\|^2$. Let

$$H(\nu) = \left[\nu\nabla^2 f(\underline{\theta}(\nu)) + \nabla^2\phi(\underline{\theta}(\nu))\right]^{-1}, \kappa = \frac{m}{M}.$$

We have

$$\frac{d}{dt}\|\theta(t) - \underline{\theta}(\nu)\|^2 = 2\left\langle \dot{\theta}(t) - \frac{\beta}{\alpha}\frac{1}{\kappa}e^{cMt/\alpha}\dot{\underline{\theta}}(\nu), \theta(t) - \underline{\theta}(\nu)\right\rangle \tag{8}$$

$$= -2\left\langle \nabla^2\phi(\theta(t))^{-1}\nabla f(\theta(t)) - \frac{\beta}{\alpha}\frac{1}{\kappa}e^{cMt/\alpha}H(\nu)\nabla f(\underline{\theta}(\nu)), \theta(t) - \underline{\theta}(\nu)\right\rangle$$

Let $G(\nu) = \frac{\beta}{\alpha}\frac{1}{\kappa}e^{cMt/\alpha}H(\nu)$. We rewrite the RHS of Equation (8) as

$$\frac{d}{dt}\|\theta(t) - \underline{\theta}(\nu)\|^2 = -2\underbrace{\left\langle \nabla^2\phi(\theta(t))^{-1}\left(\nabla f(\theta(t)) - \nabla f(\underline{\theta}(\nu))\right), \theta(t) - \underline{\theta}(\nu)\right\rangle}_{T_1}$$

$$+2\underbrace{\left\langle \left(G(\nu) - \nabla^2\phi(\theta(t))^{-1}\right)\nabla f(\underline{\theta}(\nu)), \theta(t) - \underline{\theta}(\nu)\right\rangle}_{T_2} \tag{9}$$

From strong convexity of $f$, we have the following lower bound for $T_1$

$$T_1 \geq \frac{m}{\beta}\|\theta(t) - \underline{\theta}(\nu)\|^2.$$

From Lemma 5 we have the following upper bound for $T_2$

$$T_2 \leq \|G(\nu) - \nabla^2\phi(\theta(t))^{-1}\|\|\nabla f(\underline{\theta}(\nu))\|\|\theta(t) - \underline{\theta}(\nu)\|$$

$$\leq \|G(\nu)\nabla^2\phi(\theta(t)) - I\|\|\nabla^2\phi(\theta(t))^{-1}\|\|\nabla f(\underline{\theta}(\nu))\|\|\theta(t) - \underline{\theta}(\nu)\|$$

$$\leq \frac{1}{\alpha}\left(\frac{\beta}{m\nu(t) + \beta}\right)\|\nabla f(\theta_0)\|\|G(\nu)\nabla^2\phi(\theta(t)) - I\|\|\theta(t) - \underline{\theta}(\nu)\|$$

Note that $G(\nu)\nabla^2\phi(\theta(t))$ is a PSD matrix. Its minimum and maximum eigenvalues satisfy

$$\sigma_{min}(G(\nu)\nabla^2\phi(\theta(t))) \geq \frac{c}{1 + (\frac{\alpha}{\beta}c\kappa - 1)e^{-cMt/\alpha}},$$

$$\sigma_{max}(G(\nu)\nabla^2\phi(\theta(t))) \leq \frac{\beta}{\alpha}\frac{c}{\kappa}\frac{1}{1 + (c - 1)e^{-cMt/\alpha}}.$$

Using this, we can upper bound $\|G(\nu)\nabla^2\phi(\theta(t)) - I\|$ as

$$\|G(\nu)\nabla^2\phi(\theta(t)) - I\| \leq \max\{|1 - \sigma_{min}(G(\nu)\nabla^2\phi(\theta(t)))|, |1 - \sigma_{max}(G(\nu)\nabla^2\phi(\theta(t)))|\}$$

$$\leq \max\left\{\left|1 - \frac{c}{1 + (\frac{\alpha}{\beta}c\kappa - 1)e^{-cMt/\alpha}}\right|, \frac{\beta}{\alpha}\frac{c}{\kappa}\frac{1}{1 + (c - 1)e^{-cMt/\alpha}} - 1\right\}$$

$$= \frac{\beta}{\alpha}\frac{c}{\kappa}\frac{1}{1 + (c - 1)e^{-cMt/\alpha}} - 1,$$

where the last equality follows from our choice of $c$. Letting the above upper bound be $\sigma_G$, we get the following bound for $T_2$

$$T_2 \le \frac{\sigma_G}{\alpha} \left( \frac{\beta}{m\nu(t) + \beta} \right) \|\nabla f(\theta_0)\| \|\theta(t) - \underline{\theta}(\nu)\|$$

Substituting these bounds in Equation (9) gives us

$$\frac{d}{dt} \|\theta(t) - \underline{\theta}(\nu)\|^2 \le -2\frac{m}{\beta} \|\theta(t) - \underline{\theta}(\nu)\|^2 + 2\frac{\sigma_G}{\alpha} \left( \frac{\beta}{m\nu(t) + \beta} \right) \|\nabla f(\theta_0)\| \|\theta(t) - \underline{\theta}(\nu)\|.$$

We now solve the above ODE. Let $g(t) = \|\theta(t) - \underline{\theta}(\nu)\|$. The above equation can be rewritten as

$$\frac{d}{dt} g(t)^2 \le -2\frac{m}{\beta} g(t)^2 + 2\frac{\sigma_G}{\alpha} \left( \frac{\beta}{m\nu(t) + \beta} \right) \|\nabla f(\theta_0)\| g(t)$$

Let $\kappa' = \frac{\alpha}{\beta}\kappa$. Then $\frac{\sigma_G}{\alpha} \left( \frac{\beta}{m\nu(t)+\beta} \right)$ can be rewritten as

$$\frac{\sigma_G}{\alpha} \left( \frac{\beta}{m\nu(t) + \beta} \right) = \frac{1}{\alpha} \left( \frac{c}{\kappa'} - 1 + (1-c)e^{-cMt/\alpha} \right) \frac{ce^{-cMt/\alpha}}{(1 + (c-1)e^{-cMt/\alpha})^2}$$

This gives us the following upper bound for the ODE

$$\frac{d}{dt} g(t)^2 \le -2\frac{m}{\beta} g(t)^2 + \frac{1}{\alpha} \left( \frac{c}{\kappa'} - 1 + (1-c)e^{-cMt/\alpha} \right) \frac{2ce^{-cMt/\alpha}}{(1 + (c-1)e^{-cMt/\alpha})^2} \|\nabla f(\theta_0)\| g(t)$$

Note that this is the same ODE as the one obtained in the proof of Theorem 1. Solving this gives us the following upper bound for the distance $\|\theta(t) - \underline{\theta}(\nu)\|$

$$\|\theta(t) - \underline{\theta}(\nu)\| \le \frac{\beta}{\alpha} \frac{\|\nabla f(\theta_0)\|_2}{m} \left( e^{-mt/\beta} + \frac{c}{1 - c - e^{cMt/\alpha}} \right).$$

# D   Proof of Theorem 4

Note that the regularized objective is $m + \frac{1}{\nu}$ strongly convex. So we have

$$\left( \frac{m\nu + 1}{2\nu} \right) \|\theta^* - \underline{\theta}(\nu)\|^2 \le R_n(\theta^*) - R_n(\underline{\theta}(\nu)) + \frac{1}{2\nu} \left( \|\theta^*\|^2 - \|\underline{\theta}(\nu)\|^2 \right).$$

Since $R_n$ is convex, we can upper bound $R_n(\theta^*) - R_n(\underline{\theta}(\nu))$ as $-\langle \nabla R_n(\theta^*), \underline{\theta}(\nu) - \theta^* \rangle$. Using this in the above equation gives us

$$\left( \frac{m\nu + 1}{2\nu} \right) \|\theta^* - \underline{\theta}(\nu)\|^2 \le -\langle \nabla R_n(\theta^*), \underline{\theta}(\nu) - \theta^* \rangle + \frac{1}{2\nu} \left( \|\theta^*\|^2 - \|\underline{\theta}(\nu)\|^2 \right)$$

$$\le \|\nabla R_n(\theta^*)\| \left( \|\theta^* - \underline{\theta}(\nu)\| \right) + \frac{1}{2\nu} \left( \|\theta^*\| + \|\underline{\theta}(\nu)\| \right) \left( \|\theta^* - \underline{\theta}(\nu)\| \right).$$

Simplifying the above expression, we get

$$\left( \frac{m\nu + 1}{2\nu} \right) \|\theta^* - \underline{\theta}(\nu)\| \le \|\nabla R_n(\theta^*)\| + \frac{1}{2\nu} \left( \|\theta^*\| + \|\underline{\theta}(\nu)\| \right)$$

$$\le \|\nabla R_n(\theta^*)\| + \frac{1}{2\nu} \left( 2\|\theta^*\| + \|\underline{\theta}(\nu) - \theta^*\| \right).$$

For our choice of regularization parameter $\nu$: $1/\nu \ge 2\frac{\|\nabla R_n(\theta^*)\|_2}{\|\theta^*\|}$, we can upper bound the RHS as

$$\left( \frac{m\nu + 1}{2\nu} \right) \|\theta^* - \underline{\theta}(\nu)\| \le \frac{3}{2} \frac{\|\theta^*\|}{\nu} + \frac{1}{2\nu} \|\theta^* - \underline{\theta}(\nu)\|.$$

This gives us the required bound

$$\|\theta^* - \underline{\theta}(\nu)\| \le \frac{3\|\theta^*\|}{m\nu}.$$

# E   Proof of Corollary 7

To prove the corollary we first provide high probability bounds for the smallest and largest singular values of the sample covariance matrix. To this end we utilize the results of Rudelson and Vershynin [2009] on the properties of smallest singular value of a random matrix. Suppose $Z \in \mathbb{R}^{a \times b}, a > b$ be a matrix with i.i.d sub-gaussian

random variables. And let $\sigma_{min}(Z), \sigma_{max}(Z)$ be the smallest and largest singular values of $Z$. Then Rudelson and Vershynin [2009] show that

$$\mathbb{P}\left(\sigma_{min}(Z) \leq \eta(\sqrt{a} - \sqrt{b-1})\right) \leq (C\eta)^{a-b+1} + e^{-ca}. \tag{10}$$

$$\mathbb{P}\left(\sigma_{max}(Z) \geq t\sqrt{a}\right) \leq e^{-c_0 t^2 a}, \ \forall t \geq C_0, \tag{11}$$

where $c_0, C_0, c, C$ depend on sub-gaussian moment. Using these results, we can show that the there exists a constant $c_1 > 0$ such that the smallest and largest eigenvalues of the sample covariance matrix $\hat{\Sigma}$ satisfy the following with high probability

$$m := \sigma_{min}(\hat{\Sigma}) \geq 9/10, \quad M := \sigma_{max}(\hat{\Sigma}) \leq 11/10.$$

We condition on the above event in the rest of our proof. To bound $\|\theta(t) - \theta^*\|$ we use a simple triangle inequality

$$\|\theta(t) - \theta^*\| \leq \|\theta(t) - \underline{\theta}(\nu(t))\| + \|\theta^* - \underline{\theta}(\nu(t))\|,$$

where $\nu(t) = \frac{1}{cm}\left(e^{cMt} - 1\right)$.

First note that for the setting considered in the Corollary, $\|\theta^* - \underline{\theta}(\nu)\|$ can be upper bounded as

$$\|\theta^* - \underline{\theta}(\nu)\|^2 \leq \frac{1}{(1+\nu)^2}\left[\|\theta^*\|^2 + \nu^2\sigma^2\frac{p}{n}\right].$$

This follows from Theorem 6. From Theorem 1 we have

$$\|\theta(t) - \underline{\theta}(\nu(t))\| \leq \frac{\|\nabla R_n(0)\|}{m}\left(e^{-mt} + \frac{c}{1-c-e^{cMt}}\right)$$

Combining these two bounds, we get the following bound for $\|\theta(t) - \theta^*\|$

$$\|\theta(t) - \theta^*\| \leq \frac{\|\nabla R_n(0)\|}{m}\left(e^{-mt} + \frac{c}{1-c-e^{cMt}}\right) + \frac{1}{1+\nu(t)}\sqrt{\|\theta^*\|^2 + \nu(t)^2\sigma^2\frac{p}{n}}.$$

For sufficiently large $n$, $\|\nabla R_n(0)\|$ is upper bonded by $\|\theta^*\| + \sigma\sqrt{\frac{p}{n}}$. Substituting the values of $m, M$ and $\|\nabla R_n(0)\|$ in the above bound gives us the requires bound for $\|\theta(t) - \theta^*\|$.

Note that the upper bound of $\|\theta(t) - \theta^*\|$ is dominated by $\|\theta^* - \underline{\theta}(\nu(t))\|$. So it suffices to find a $t$ which minimizes this term. And it is minimized at $\nu(t) = \frac{\|\theta^*\|^2}{\sigma^2}\frac{n}{p}$. The $t$ corresponding to this $\nu(t)$ is given by

$$t = \frac{100}{99}\log\left(1 + \frac{81}{100}\frac{\|\theta^*\|^2}{\sigma^2}\frac{n}{p}\right).$$

The bound at this value of $t$ can be shown to be

$$(1+\epsilon)\left[\frac{\|\theta^*\|^2}{\|\theta^*\|^2 + \frac{\sigma^2 p}{n}}\right]\frac{\sigma^2 p}{n},$$

for some small positive constant $\epsilon$ less than 0.1.

# F  Proofs of Auxiliary Lemmas

## F.1  Proof of Lemma 3

**Part 1.**  Let $h(t) = \|\nabla f(\theta(t))\|_2^2$. Then

$$\frac{d}{dt}h(t) = 2\left\langle\nabla^2 f(\theta(t))\dot{\theta}(t), \nabla f(\theta(t))\right\rangle = -2\left\langle\nabla^2 f(\theta(t))\nabla f(\theta(t)), \nabla f(\theta(t))\right\rangle$$

$$\leq -2m\|\nabla f(\theta(t))\|_2^2 = -2mh(t).$$

Letting $g(t) = \log h(t)$ we can rewrite the above equation as

$$\frac{d}{dt}g(t) \leq -2m.$$

Integrating the LHS and RHS of the above expression gives us $\|\nabla f(\theta(t))\|_2^2 \leq e^{-2mt}\|\nabla f(\theta_0)\|_2^2$.

**Part 2.** Let

$$H(\nu) = \left[\nu\nabla^2 f(\underline{\theta}(\nu)) + I\right]^{-1}, \kappa = \frac{m}{M}.$$

Let $h(\nu) = \|\nabla f(\underline{\theta}(\nu))\|_2^2$. Then

$$\frac{d}{d\nu}h(\nu) = 2\left\langle \nabla^2 f(\underline{\theta}(\nu))\dot{\underline{\theta}}(\nu), \nabla f(\underline{\theta}(\nu))\right\rangle = -2\left\langle \nabla^2 f(\underline{\theta}(\nu))H(\nu)\nabla f(\underline{\theta}(\nu)), \nabla f(\underline{\theta}(\nu))\right\rangle.$$

Since the smallest eigen value of $\nabla^2 f(\underline{\theta}(\nu))H(\nu)$ is equal to $\frac{m}{m\nu+1}$, we have

$$\frac{d}{d\nu}h(\nu) \le -\frac{2m}{m\nu+1}\|\nabla f(\underline{\theta}(\nu))\|_2^2 = -\frac{2m}{m\nu+1}h(\nu).$$

Letting $g(\nu) = \log h(\nu)$ we can rewrite the above equation as

$$\frac{d}{d\nu}g(\nu) \le -\frac{2m}{m\nu+1}.$$

This gives us the following bound for $\|\nabla f(\underline{\theta}(\nu))\|^2$

$$\|\nabla f(\underline{\theta}(\nu))\|^2 \le \frac{1}{(m\nu+1)^2}\|\nabla f(\theta_0)\|^2$$

## F.2 Proof of Lemma 4

**Part 1.** The first part of the Lemma follows from the following two inequalities

$$f(\hat{\theta}) \le f(\underline{\theta}(\nu)),$$

$$f(\underline{\theta}(\nu)) + \frac{1}{\nu}D_\phi(\underline{\theta}(\nu), \theta_0) \le f(\hat{\theta}) + \frac{1}{\nu}D_\phi(\hat{\theta}, \theta_0).$$

Combining these two inequalities shows that $D_\phi(\underline{\theta}(\nu), \theta_0) \le D_\phi(\hat{\theta}, \theta_0)$.

**Part 2.** To prove the second part we show that $\frac{d}{dt}D_\phi(\hat{\theta}, \theta(t)) \le 0$. This will show that for any $t \in [0, \infty)$

$$D_\phi(\hat{\theta}, \theta(t)) \le D_\phi(\hat{\theta}, \theta_0).$$

$\frac{d}{dt}D_\phi(\hat{\theta}, \theta(t))$ can be written as

$$\frac{d}{dt}D_\phi(\hat{\theta}, \theta(t)) = \frac{d}{dt}\left(\phi(\hat{\theta}) - \phi(\theta(t)) - \left\langle \nabla\phi(\theta(t)), \hat{\theta} - \theta(t)\right\rangle\right)$$

$$= -\left\langle \nabla^2\phi(\theta(t))\dot{\theta}(t), \hat{\theta} - \theta(t)\right\rangle$$

$$= -\left\langle \nabla f(\hat{\theta}) - \nabla f(\theta(t)), \hat{\theta} - \theta(t)\right\rangle \le 0.$$

## F.3 Proof of Lemma 5

Let

$$H(\nu) = \left[\nu\nabla^2 f(\underline{\theta}(\nu)) + \nabla^2\phi(\underline{\theta}(\nu))\right]^{-1}, \kappa = \frac{m}{M}.$$

Let $h(\nu) = \|\nabla f(\underline{\theta}(\nu))\|_2^2$. Then

$$\frac{d}{d\nu}h(\nu) = 2\left\langle \nabla^2 f(\underline{\theta}(\nu))\dot{\underline{\theta}}(\nu), \nabla f(\underline{\theta}(\nu))\right\rangle = -2\left\langle \nabla^2 f(\underline{\theta}(\nu))H(\nu)\nabla f(\underline{\theta}(\nu)), \nabla f(\underline{\theta}(\nu))\right\rangle.$$

Since the smallest eigen value of $\nabla^2 f(\underline{\theta}(\nu))H(\nu)$ is greater than or equal to $\frac{m}{m\nu+\beta}$, we have

$$\frac{d}{d\nu}h(\nu) \le -\frac{2m}{m\nu+\beta}\|\nabla f(\underline{\theta}(\nu))\|_2^2 = -\frac{2m}{m\nu+\beta}h(\nu).$$

Letting $g(\nu) = \log h(\nu)$ we can rewrite the above equation as

$$\frac{d}{d\nu}g(\nu) \le -\frac{2m}{m\nu+\beta}.$$

This gives us the following bound for $\|\nabla f(\underline{\theta}(\nu))\|^2$

$$\|\nabla f(\underline{\theta}(\nu))\|^2 \le \left(\frac{\beta}{m\nu+\beta}\right)^2 \|\nabla f(\theta_0)\|^2.$$

# G   Proof of Theorem 8

Let $\hat{\theta}$ be the minimizer of $f(\theta)$ which is closest to $\theta_0$. Since $\underline{\theta}(\nu)$ is the minimizer of the regularized objective, we have

$$f(\underline{\theta}(\nu)) + \frac{1}{2\nu}\|\underline{\theta}(\nu) - \theta_0\|_2^2 \le f(\hat{\theta}) + \frac{1}{2\nu}\|\hat{\theta} - \theta_0\|_2^2$$

Moreover, since $\hat{\theta}$ is a minimizer of $f$, we have

$$f(\hat{\theta}) \le f(\underline{\theta}(\nu)).$$

Combining the two equations, we get

$$\forall \nu \quad f(\hat{\theta}) \le f(\underline{\theta}(\nu)) \le f(\hat{\theta}) + \frac{1}{2\nu}\|\hat{\theta} - \theta_0\|_2^2.$$

This shows that $\lim_{\nu \to \infty} f(\underline{\theta}(\nu)) = f(\hat{\theta})$. We now show that $\lim_{\nu \to \infty} \underline{\theta}(\nu) = \hat{\theta}$. We do this in 3 stages.

1. $\underline{\theta}(\nu)$ **converges.** We first show that the sequence $\{\underline{\theta}(\nu)\}_\nu$ converges. Suppose the sequence diverges. It means that for any $\tilde{\theta} \in \mathbb{R}^p$, the following is true

$$\exists \epsilon, \forall \nu_0, \exists \nu > \nu_0, \text{ such that } \|\underline{\theta}(\nu) - \tilde{\theta}\| > \epsilon.$$

   That is, for any $\tilde{\theta}$ we can find a sequence of $\nu$'s going to infinity such that $\|\underline{\theta}(\nu) - \tilde{\theta}\| > \epsilon$ for any $\nu$ in the sequence. We now choose $\tilde{\theta} = \hat{\theta}$.

   - **Case 1.** Suppose $f(\underline{\theta}(\nu)) = f(\hat{\theta})$ for some $\nu$ in the sequence. Since $\hat{\theta}$ is the closest minimizer to $\theta_0$ and since $\|\underline{\theta}(\nu) - \hat{\theta}\| > \epsilon$, $\underline{\theta}(\nu)$ can't be the minimizer of the regularized objective. So this case can never happen.
   - **Case 2.** We have a sequence of $\nu$'s going to infinity such that $f(\underline{\theta}(\nu)) > f(\hat{\theta})$. This again contradicts the fact that $\underline{\theta}(\nu)$ is the minimizer of the regularized objective.

   This shows that the sequence $\{\underline{\theta}(\nu)\}_\nu$ always converges.

2. $\underline{\theta}(\nu)$ **converges to a minimizer.** Suppose $\underline{\theta}(\nu)$ converges to a point $\theta^*$ which is a not a minimizer. Since $\underline{\theta}(\nu)$ is the minimizer of the regularized objective, we have

$$\forall \nu, \quad \left(f(\underline{\theta}(\nu)) + \frac{1}{2\nu}\|\underline{\theta}(\nu) - \theta_0\|_2^2\right) - \left(f(\hat{\theta}) + \frac{1}{2\nu}\|\hat{\theta} - \theta_0\|_2^2\right) \le 0.$$

   This shows that

$$\lim_{\nu \to \infty} f(\underline{\theta}(\nu)) + \frac{1}{2\nu}\|\underline{\theta}(\nu) - \theta_0\|_2^2 - f(\hat{\theta}) - \frac{1}{2\nu}\|\hat{\theta} - \theta_0\|_2^2 \le 0.$$

   Computing the limit in the LHS of the above expression we get

$$f(\theta^*) - f(\hat{\theta}) \le 0.$$

   This contradicts our initial assumption that $\underline{\theta}(\nu)$ doesn't converge to a minimizer.

3. $\underline{\theta}(\nu)$ **converges to the minimizer closest to $\theta_0$.** Lets suppose $\underline{\theta}(\nu)$ converges to a minimizer $\theta^*$ such that

$$\|\theta^* - \theta_0\| > \|\hat{\theta} - \theta_0\|.$$

   Let $c = \|\theta^* - \theta_0\| - \|\hat{\theta} - \theta_0\|$. From the definition of convergence we know that
$$\forall \epsilon, \exists \nu_0 \text{ such that } \forall \nu > \nu_0, \|\underline{\theta}(\nu) - \theta^*\| \le \epsilon.$$

   Fix any $\epsilon$. Let $\nu_0$ be as defined in the above definitions of convergence. For any $\nu > \nu_0$ we have

$$f(\underline{\theta}(\nu)) + \frac{1}{2\nu}\|\underline{\theta}(\nu) - \theta_0\|_2^2 \ge f(\theta^*) + \frac{1}{2\nu}\left(\|\theta^* - \theta_0\|_2^2 + \|\underline{\theta}(\nu) - \theta^*\|_2^2 - 2\|\theta^* - \theta_0\|\|\underline{\theta}(\nu) - \theta^*\|\right),$$

   where the inequality follows from triangle inequality $\|\underline{\theta}(\nu) - \theta_0\|_2 \ge \|\theta^* - \theta_0\|_2 - \|\underline{\theta}(\nu) - \theta^*\|_2$ and the fact that $f(\theta^*) \le f(\underline{\theta}(\nu))$.

   Rearranging the terms in the RHS of the above expression and using the fact that $\|\underline{\theta}(\nu) - \theta^*\|_2 \le \epsilon$, we get

$$f(\underline{\theta}(\nu)) + \frac{1}{2\nu}\|\underline{\theta}(\nu) - \theta_0\|_2^2 \ge f(\theta^*) + \frac{1}{2\nu}\|\theta^* - \theta_0\|_2^2 - \frac{1}{2\nu}\left(\epsilon^2 + 2\epsilon\|\theta^* - \theta_0\|\right).$$

   Replacing $\|\theta^* - \theta_0\|$ with $c + \|\hat{\theta} - \theta_0\|$, we get

$$f(\underline{\theta}(\nu)) + \frac{1}{2\nu}\|\underline{\theta}(\nu) - \theta_0\|_2^2 \ge f(\hat{\theta}) + \frac{1}{2\nu}\|\hat{\theta} - \theta_0\|_2^2 \\ + \frac{1}{2\nu}\left(c^2 - \epsilon^2 + 2(c - \epsilon)\|\hat{\theta} - \theta_0\| - 2c\epsilon\right). \tag{12}$$

   Choose any $\epsilon < \frac{c}{100}$. Then the above equation shows that there exists a $\nu$ such that

$$f(\underline{\theta}(\nu)) + \frac{1}{2\nu}\|\underline{\theta}(\nu) - \theta_0\|_2^2 > f(\hat{\theta}) + \frac{1}{2\nu}\|\hat{\theta} - \theta_0\|_2^2.$$

   However, this can't happen because $\underline{\theta}(\nu)$ is the minimizer of the regularized objective. So this contradicts our initial assumption that $\underline{\theta}(\nu)$ doesn't converge to the minimizer that is closest to $\theta_0$.

# H   Proof of Theorem 9

## H.1   Preliminaries

For clarity of analysis, we follow our continuous time approach of previous sections. From first order optimality of $\underline{\theta}(\nu)$, we have that

$$\frac{\underline{\theta}(\nu)}{\nu} + \nabla R_n(\underline{\theta}(\nu)) = 0 \tag{13}$$

Using this, firstly, we list the derivatives of $R_n(\underline{\theta}(\nu))$, $\|\underline{\theta}(\nu)\|_2$ and $\underline{\theta}(\nu)$ w.r.t. $\nu$.

- The derivative of $\underline{\theta}(\nu)$ can be written as:

$$\dot{\underline{\theta}(\nu)} = -\left[\mathcal{I}_p + \nu\nabla^2 R_n(\underline{\theta}(\nu))\right]^{-1}(\nabla R_n(\underline{\theta}(\nu))) \tag{14}$$

- 

$$\dot{R}_n(\underline{\theta}(\nu)) = -\nabla R_n(\underline{\theta}(\nu))^T \left[\mathcal{I}_p + \nu\nabla^2 R_n(\underline{\theta}(\nu))\right]^{-1} \nabla R_n(\underline{\theta}(\nu)) \tag{15}$$

- 

$$\frac{d\|\underline{\theta}(\nu)\|_2}{d\nu} = \frac{\dot{\underline{\theta}(\nu)}^T \underline{\theta}(\nu)}{\|\underline{\theta}(\nu)\|_2} = \frac{\underline{\theta}(\nu)^T \left[\mathcal{I}_p + \nu\nabla^2 R_n(\underline{\theta}(\nu))\right]^{-1} \underline{\theta}(\nu)}{\nu\|\underline{\theta}(\nu)\|_2} \tag{16}$$

Next, we list down some properties of $R_n(\theta)$. Recall that $\phi(y\theta^T x) = \exp(-\theta^T(yx))$. For brevity, let $x_i y_i = z_i$. Then we have the following:

- 
  **Claim 1.** $\lambda_{\max}\left(\nabla^2 R_n(\underline{\theta}(\nu))\right) \leq R_n(\underline{\theta}(\nu))$

  To see this, for any unit vector $v$, we have that

  $$v^T \nabla^2 R_n(\underline{\theta}(\nu))v = \frac{1}{n}\sum_{i=1}^{n}(z_i^T v)^2 \exp(-\theta^T z_i) \leq \left(\max_i (z_i^T v)^2\right)\left(\frac{1}{n}\sum_{i=1}^{n}\exp(-\theta^T z_i)\right) \leq R_n(\underline{\theta}(\nu)),$$

  where we have used that $\|x_i y_i\|_2 \leq 1$.

- 
  **Claim 2.** $\lambda_{\min}\left(\nabla^2 R_n(\underline{\theta}(\nu))\right) \geq \frac{1}{n}\exp\left(-\|\underline{\theta}(\nu)\|_2\right)$

  To see this, for any unit vector $v$, we have that

  $$v^T \nabla^2 R_n(\underline{\theta}(\nu))v = \frac{1}{n}\sum_{i=1}^{n}(z_i^T v)^2 \exp(-\theta^T z_i)$$

  where we have used that $\|x_i y_i\|_2 \leq 1$.

We also make use of the following duality:

**Lemma 6.** *[Lemma 2 [Nacson et al., 2018]] The following duality holds:*

$$\frac{\|\nabla R_n(\underline{\theta}(\nu))\|_2}{R_n(\theta)} \geq \max_{\|w\|_2=1} \min_{i \in n} w^T(y_i x_i)$$

*Proof.* The proof follows from convex duality arguments and can be found in [Nacson et al., 2018]. □

Note that by our assumptions of the max-margin being 1, we have that $\|\nabla R_n(\underline{\theta}(\nu))\|_2 \geq R_n(\theta)$.

**Lemma 7.** *[Hoorfar and Hassani, 2008] The product-log or the Lambert-W Function satisfies,*

$$\log\nu - \log\log\nu + \frac{\log\log\nu}{2\log\nu} \leq \mathcal{W}(\nu) \leq \log\nu - \log\log\nu + \frac{e}{e-1}\frac{\log\log\nu}{\log\nu}$$

### H.2 Main Proof

- We begin by proving bounds on $R_n(\underline{\theta}(\nu))$. We can upper bound the $\dot{R}_n(\underline{\theta}(\nu))$ as:

$$\dot{R}_n(\underline{\theta}(\nu)) = -\nabla R_n(\underline{\theta}(\nu))^T \left[\mathcal{I}_p + \nu\nabla^2 R_n(\underline{\theta}(\nu))\right]^{-1} \nabla R_n(\underline{\theta}(\nu)) \tag{17}$$

  Moreover, from Claim 1, we have that

$$\nabla R_n(\underline{\theta}(\nu))^T \left[\mathcal{I}_p + \nu\nabla^2 R_n(\underline{\theta}(\nu))\right]^{-1} \nabla R_n(\underline{\theta}(\nu)) \geq \lambda_{\min}\left(\left[\mathcal{I}_p + \nu\nabla^2 R_n(\underline{\theta}(\nu))\right]^{-1}\right) \|\nabla R_n(\underline{\theta}(\nu))\|_2^2 \tag{18}$$

$$\geq \frac{1}{1 + \nu R_n(\underline{\theta}(\nu))} \|\nabla R_n(\underline{\theta}(\nu))\|_2^2 \tag{19}$$

  Further, using Lemma 6 we have that

$$\nabla R_n(\underline{\theta}(\nu))^T \left[\mathcal{I}_p + \nu\nabla^2 R_n(\underline{\theta}(\nu))\right]^{-1} \nabla R_n(\underline{\theta}(\nu)) \geq \frac{\gamma^2 R_n^2(\underline{\theta}(\nu))}{1 + \nu R_n(\underline{\theta}(\nu))} \tag{20}$$

  Combining (15) and (20), we get,

$$\dot{R}_n(\underline{\theta}(\nu)) \leq \frac{-R_n^2(\underline{\theta}(\nu))}{1 + \nu R_n(\underline{\theta}(\nu))}$$

  Solving the above ODE, we get that

$$R_n(\underline{\theta}(\nu)) \leq \frac{\mathcal{W}(\nu c_1)}{\nu}, \tag{21}$$

  where $\mathcal{W}$ is the product-log function. This proves the first claim of the Theorem.

- Now, we prove bounds on $\|\underline{\theta}(\nu)\|_2$.

$$\frac{d\|\underline{\theta}(\nu)\|_2}{d\nu} = \frac{\underline{\theta}(\nu)^T \left[\mathcal{I}_p + \nu\nabla^2 R_n(\underline{\theta}(\nu))\right]^{-1} \underline{\theta}(\nu)}{\nu\|\underline{\theta}(\nu)\|_2} \geq \lambda_{\min}\left(\left[\mathcal{I}_p + \nu\nabla^2 R_n(\underline{\theta}(\nu))\right]^{-1}\right)\frac{\|\underline{\theta}(\nu)\|_2}{\nu} \tag{22}$$

$$= \frac{1}{1 + \nu\lambda_{\max}(\nabla^2 R_n(\underline{\theta}(\nu)))}\frac{\|\underline{\theta}(\nu)\|_2}{\nu} \tag{23}$$

  Using the upper bound on the loss and Claim 1, we get that,

$$\frac{d\|\underline{\theta}(\nu)\|_2}{d\nu} \geq \frac{1}{1 + \nu\mathcal{W}(\nu)}\frac{\|\underline{\theta}(\nu)\|_2}{2}$$

  Solving the above ODE, we get that

$$\|\theta_\nu\|_2 \geq \mathcal{W}(\nu)$$

  This proves the lower bound on $\|\theta\|_2$. To prove the upper bound, we follow the same procedure as above along with Claim 2. This gives us:

$$\frac{d\|\underline{\theta}(\nu)\|_2}{d\nu} = \frac{\underline{\theta}(\nu)^T \left[\mathcal{I}_p + \nu\nabla^2 R_n(\underline{\theta}(\nu))\right]^{-1} \underline{\theta}(\nu)}{\nu\|\underline{\theta}(\nu)\|_2} \leq \lambda_{\max}\left(\left[\mathcal{I}_p + \nu\nabla^2 R_n(\underline{\theta}(\nu))\right]^{-1}\right)\frac{\|\underline{\theta}(\nu)\|_2}{\nu} \tag{24}$$

$$= \frac{1}{1 + \nu\lambda_{\min}(\nabla^2 R_n(\underline{\theta}(\nu)))}\frac{\|\underline{\theta}(\nu)\|_2}{\nu} \tag{25}$$

$$= \frac{1}{1 + \nu\lambda_{\min}(\frac{1}{n}\exp\left(-\|\theta\|_2\right))}\frac{\|\underline{\theta}(\nu)\|_2}{\nu} \tag{26}$$

  Solving the ODE, we get that:

$$\|\theta\|_2 \leq \mathcal{W}\left(\frac{\nu}{n}\exp(c_1\nu/n)\right) - c_1\nu/n.$$

  Combining the upper and lower bound, we get that for large enough $t$, $\|\underline{\theta}(\nu)\|_2 \approx \log\nu$.

- Now, to prove the bound on the margin, observe that

$$\max_i \exp(-\underline{\theta}(\nu)^T z_i) \leq nR_n(\underline{\theta}(\nu)) \leq \frac{n\mathcal{W}(\nu)}{\nu}$$

  Taking $-\log(\cdot)$ on both sides and using that $\|\underline{\theta}(\nu)\|_2 \approx \log\nu$, we get that

$$\frac{\min_i y_i\underline{\theta}(\nu)^T x_i}{\|\underline{\theta}(\nu)\|_2} \geq 1 - \frac{\log n + \log(W(\nu))}{\log(\nu)} \approx 1 - \frac{\log\log(\nu)}{\log\nu} - \frac{\log n}{\log\nu}$$

This completes the proof of the theorem.

### H.3 Proof of Lemma 1

Consider the following 2-D function $f : \mathbb{R} \times (-100, \infty) \mapsto \mathbb{R}$,

$$f(x, y) = \frac{(x+1)^2}{y + 100}$$

First note that this function is convex. In particular, for $y > -100$, the hessian is given by,

$$\nabla^2 f = \begin{bmatrix} \frac{2}{y+100} & \frac{-2(x+1)}{(y+100)^2} \\ \frac{-2(x+1)}{(y+100)^2} & \frac{2(x+1)^2}{(y+100)^3} \end{bmatrix} \tag{27}$$

Some algebra shows that the eigenvalues for the hessian above are given by $(0, \frac{2}{(y+100)} + 2\frac{(x+1)^2}{(y+100)^3})$, which are non-negative for $y > -100$.

For this function, we now show that the regularization and optimization paths don't converge to the same point. In particular, suppose $(x_0, y_0) = (2, 1)$. For this, we know that the regularization path converges to the nearest minimizer $(-1, 1)$, whereas the continuous time gradient descent converges to $(-1, 1.0223)$. The reason why continuous time GD doesn't converge to the nearest minimizer is that the gradient at any point has a non zero component along $y$-axis.

## Footnotes

[1]To very the above bound, one can plugin the bound in the ODE.