[Reviews · NeurIPS 2018]

Reviewer 1



The authors explore the relation between the trajectory of Gradient Descent (GD) initiated in the origin and the regularization path for l2-regularized minimization of the same objective. They first study the continuous-time setting where GD is replaced Gradient Flow, assuming that the objective is smooth and strongly convex. The main result (Theorem 1 whose proof I have verified) is as follows: under the appropriate scaling between the time $t$ in GD and the inverse regularization parameter $\eta$, the two trajectories do not diverge much. This result is obtained by quantifying the shrinkage of the gradients as $t$ and $eta$ tend to infinity. In the continuous-time setting, the authors manage to reduce this task to formulating and solving certain ODEs. Then, a discrete-time counterpart of Theorem 1 is established (Theorem 2); its proof is more technical but mostly goes along the same lines (the main technical difference is that instead of solving ODEs one deals with linear recursions). Honestly, I did not check the proof of this later result, but overall it seems credible to me taking into account my verification of Theorem 1 and related technical lemmas. The authors then briefly state the extensions of these results for non-Euclidean proximal setups (i.e. Mirror Descent), assuming that the potential is smooth and strongly convex. Finally, the obtained results are applied to establish statistical bounds for the excess risk when Gradient Flow is used to minimize the empirical risk objective. The main result here is Corollary 5, which provides a recipe for early stopping. These results are then tightened for random-design linear regression and classification under separability assumptions. These results are close in spirit to (1) the results of (Hardt et al, 2015), where implicit regularization properties of SGD with early stopping are established in the general setting of empirical risk minimization (ERM), and (2) a more recent line of work (Soudry et al. 2017), (Ji and Telgarsky 2018), (Gunasekar et a;. 2018), where self-regularization properties of SGD are established in the case of separable data. First let me discuss the main highlight of the paper, Theorems 1-2. The technique that relates the two trajectories through a differential equation is easy-to-use and powerful, and I consider it the main highlight of the paper. I also like the idea of using the known results for l2-regularized ERM to analyze GD with early stopping. That said, it has long been known that the analysis of Gradient Flow is reduced to solving ODEs, see e.g. (D. Scieur, Integration Methods and Accelerated Optimization Algorithms 2017). I would like to see a review of the related literature since I suspect that some of the results in Section 1 can be already known; as far as I can see, the real novelties here are (1) using the ODE-analysis to relate Gradient Flow/Descent to the l2-regularization path, and (2) providing a result for the case of discrete step. I should also mention that I find the results stated in Sections 3 and 4 particularly interesting and useful for the computational learning theory community. That said, a more thorough discussion of these results and comparison with the existing literature on optimal stopping / margin-type results would be appreciated. I have decided to evaluate the paper only as weak accept, for a number of reasons listed below: 1. On several occasions, mathematical results are claimed without a proof or reference (lines 69-70, line 80, lines 134-137). 2. The last (non-trivial) step in the proof of Theorem 1 is missing — the authors do not detail how they solve the ODE in the display between l.337 and l.338. I did manage to arrive at l.339 after some algebraic manipulations, making the estimate m\nu + 1 \ge m\nu, and solving an inhomogeneous ODE obtained in the end; I think that all these steps should have been at least hinted on! 3. Theorem 2 is not properly commented upon. 4. The results in Section 2.1 seem rather conceptual: the only known to me non-trivial case of a strongly convex *and* smooth potential is that of Mahalanobis distance which can be reduced to the case of Euclidean norm and |.|^2/2-potential by a linear change of variables. As noted by the authors themselves, the truly’’ non-Euclidean setups, such as minimization on the simplex or l1-ball, or on the nuclear-norm ball, do not admit smooth potentials, see e.g. (Nesterov and Nemirovski, On first-­order algorithms for l 1/nuclear norm minimization). Instead, one imposes a weaker property of the slow growth of the potential on the feasible set. Most importantly, as noted in (Nesterov 2005) and also by the authors themselves, non-Euclidean setups are never better than the Euclidean one when the objective is both smooth and strongly convex. 5. The quality of writing in general could be improved. For example, on several occasions I spotted contractions in the text, such as “till” or “don’t”; I personally consider their use inappropriate in a scientific paper. 6. The plots are hard to read because of very small axis tick labels and non-uniform grid. If the paper gets accepted, I would appreciate better-looking plots in the camera-ready version, as well as more ample explanations for them.

Reviewer 2



The paper establishes connections between the regularization path and optimization path of first-order methods for empirical risk minimization. This is established for both continous and discrete gradient descent and mirror descent. Finally the morivation for doing so is established via providing excess risk bounds. The connection seem to be novel to my knowledge but at the same time seems intuitive once reading the statement of the theorem. But the consequences of this connection to excess risk is not clear. Are there any other consequences ? According to the rates given in line 182, if 't' is picked as 1/m* log(sqrt{n/p}*sigma^{-1}), then the overall bound is still O(sigma* \sqrt{p/n}). This choice of 't' is almost same as the one in line 137, in terms of dependence on 'n'. Can the authors elaborate on the advantages of the proposed approach specifically with respect to early-stopping ? I went over the proofs and they seem correct to me. In the current form, this looks like a direction worth exploring but the results in the current draft are rather preliminary.

Reviewer 3



In this paper, the authors studied a connection between optimization and regularization paths. The main results show that the iterates derived by gradient descent are close to the solution of regularization paths if the time index and regularization parameters are appropriately selected. This relationship is then used to study generalization analysis of gradient descent. (1) It seems that the derived bound $O(e^{-mt}-e^{-Mt})$ shows not essential advantages over the bound $O(e^{-mt}+e^{-Mt})$ based on traditional approaches. Indeed, both the bounds are of the order $O(e^{-mt})$ if $m\neq M$. (2) In Theorem 4, it is required that the empirical risk $R_n(\theta)$ is strongly convex, which is a bit restrict. In my opinion, the empirical risk generally can not be strongly convex. Also, the assumption $1/v\geq 2\|\nabla R_n(\theta)^*\|/\|\theta^*\|$ is not natural enough since the right-hand side is a random variable. (3) Corollary 5 requires $t$ to be upper bounded by a random variable. Perhaps we are more interested in the case of $t$ large than a constant. Moreover, the derived bound on $\|\theta(t)-\theta^*\|$ in the last inequality of Section 3.1 is of the same order of the term based on traditional error decomposition, as stated in the last paragraph of page 7. If this is the case, the analysis in this paper shows no advantage over traditional analysis. (4) In the comparison with stability, the authors claimed that their approach can obtain high-probability statements. However, this seems not to be true. Indeed, with a restatement, the bound grows as a square-root function of $1/\delta$ rather than a logarithmic function of $1/\delta$, where $\delta$ is the confidence parameter. Furthermore, the authors claim the bound based on stability does not provide an early stopping rule. I am afraid this is not the case. The bound in line 176 suggests that one can stop the optimization if $e^{-mt}=O(1/n)$. (5) In the experiments with classification, the sample size is a bit too small.